# Study on In-Situ Tool Wear Detection during Micro End Milling Based on Machine Vision

**DOI:** 10.3390/mi14010100

**Published:** 2022-12-30

**Authors:** Xianghui Zhang, Haoyang Yu, Chengchao Li, Zhanjiang Yu, Jinkai Xu, Yiquan Li, Huadong Yu

**Affiliations:** 1Ministry of Education Key Laboratory for Cross-Scale Micro and Nano Manufacturing, Changchun University of Science and Technology, Changchun 130022, China; 2School of Mechanical and Aerospace Engineering, Jilin University, Changchun 130025, China

**Keywords:** micro milling, tool wear, in-situ detection, machine vision, image processing, microscopic images

## Abstract

Most in situ tool wear monitoring methods during micro end milling rely on signals captured from the machining process to evaluate tool wear behavior; accurate positioning in the tool wear region and direct measurement of the level of wear are difficult to achieve. In this paper, an in situ monitoring system based on machine vision is designed and established to monitor tool wear behavior in micro end milling of titanium alloy Ti6Al4V. Meanwhile, types of tool wear zones during micro end milling are discussed and analyzed to obtain indicators for evaluating wear behavior. Aiming to measure such indicators, this study proposes image processing algorithms. Furthermore, the accuracy and reliability of these algorithms are verified by processing the template image of tool wear gathered during the experiment. Finally, a micro end milling experiment is performed with the verified micro end milling tool and the main wear type of the tool is understood via in-situ tool wear detection. Analyzing the measurement results of evaluation indicators of wear behavior shows the relationship between the level of wear and varying cutting time; it also gives the main influencing reasons that cause the change in each wear evaluation indicator.

## 1. Introduction

Micro milling is one of the common micro cutting methods, and tools used in the machining operation are relatively small. The range of the diameter of such cutting tools has not been definitive at present, but milling tools with a typical diameter between 0.1 mm and 1 mm are widely used [1,2,3]. During the micro milling process, tool wear is an unfavorable process that directly affects both the surface quality of machined parts and machining accuracy. Excessive wear or tool damage can also cause catastrophic tool failure, resulting in scrapped parts, and even machine tool damage and accidents. Obviously, it is critical to understand tool wear behavior to determine the correct time to change a tool or shut down operations in the quest to achieve machining accuracy and efficiency. Generally, during micro machining, operators rely on sights, sounds, vibration and machining conditions to indicate when a tool needs to be replaced. As tools used in micro milling operations, however, are relatively small, the above signals such as acoustic vibration and noise are not easily detectible to help operators effectively understand wear behavior. Thus, a situation can be encountered where tools fail to apply as many times as possible before discarding, and hence machine tool shutdown time is extended due to frequent replacement of milling, thereby reducing the production efficiency. More importantly, subjective assessment of tool life is inconsistent with automation and intelligence of numerically controlled machine tools. On the other hand, quantitative estimation of the wear amount, such as wear width and wear area, is necessary during precision or ultra-precision machining process for exploring the relationship between tool wear and quality of machined surface and machining parameters or improving machining accuracy using error compensation techniques. Therefore, tool wear detection technology has always been a hotspot for researchers all over the world.

Nowadays, in situ tool wear-monitoring technology is still in a research stage. Only a few tool wear monitoring systems have been applied to machine tools. One of the main detection methods is the indirect measurement method, the principle of which is to collect the signal characteristics related to the tool wear state through sensors and extract the effective signal parameters through signal processing, and finally monitor the tool wear state through a judgment system; the types of signals collected include: current signal, sound signal, acoustic emission (AE) signal, cutting force signal or vibration signal, etc. Yang et al. [4] proposed a CNN network-based tool wear status monitoring method to solve the problem of tool wear status monitoring under complex multiple working conditions. Huang, He-Xiang et al. [5] processed the vibration signal with a 3-KMBS algorithm to predict tool wear. Zhuang et al. [6] fused vibration signals, cutting temperature signals and cutting force signals to propose a tool wear status monitoring method based on the data-driven approach. Prakash et al. [7] used acoustic emission signals to identify the wear state of micro milling tools to investigate the relationship between acoustic emission signals and the chip formation mechanism. This type of indirect measurement method based on cutting signal acquisition and processing enables monitoring the condition of the tool during cutting in addition to predicting the tool wear status. However, such methods require the use of professional signal acquisition equipment, bringing high economic costs to build the system; at the same time, to ensure the prediction accuracy also requires the acquisition of a large number of cutting signal data to develop an effective signal processing algorithm, resulting in increased difficulty in algorithm development, resulting in higher time costs.

In consideration of the above-mentioned problems faced by indirect monitoring methods, tool wear monitoring methods based on machine vision have been expanding in recent years. Thanks to its high reliability, high precision and non-contact measurement, this method can be used to monitor tool wear directly. Thus, the tool wear in situ detection method based on machine vision has attracted a lot of attention from researchers in recent years. Jia Binghui et al. [8]. established an in situ tool wear detection system based on machine vision and proposed an interactive defect extraction algorithm by searching in an 8-connected neighborhood through analysis of gray scale distribution in the wear zone. Dai et al. [9]. conducted an in situ detection on the wear behavior of a micro milling cutter having a diameter of 0.8 mm using a self-designed machine vision system. As to the obvious wear zone and higher brightness, a projection method was employed to measure parameters of the milling tool. The experiment showed that the self-designed machine vision system could reflect the trends in tool life. Hou et al. [10]. performed in situ tool wear monitoring on a milling cutter having a diameter of 20 mm. The least-squares method was used to rebuild the boundary of the tool, since no apparent wear zone was detected, and the maximum wear width was obtained. Li et al. [11] established an automatic tool wear image detection system to segment the wear images by finding the optimal threshold value through maximum inter-class variance and iteration, and counted the wear interval pixels to calculate the tool wear quantization value. Lin et al. [12] used machine vision and machine learning based methods to segment and identify tool wear areas and have achieved detection of tool wear status.

It is seen that the above-mentioned milling tools involved in tool wear in situ detection based on machine vision are mostly for conventional size tools, having a diameter from 5 mm to 10 mm, but there is less application for micro milling tools, monitoring tool wear of milling cutters, with a diameter lower than 5 mm or 1 mm. Since there is no unified wear evaluation standard for micro milling tools, and the application of micro milling has become progressively more widespread, research work in this area is needed.

For the purpose of in situ tool wear detection during the micro end milling process, this work began with analyzing types of tool wear zone, and proper indicators were identified to evaluate tool wear based on tool wear characteristics before a tool wear in situ monitoring system was built. Next, in this work, we proposed a tool wear image processing algorithm aiming to (1) improve the quality of images by removing noise from images captured during micro end milling process; (2) separate the Region of Interest (ROI) of tool wear images from the background by segmenting tool wear images through a tool wear image segmentation algorithm based on the region growing method; (3) rotate and locate ROI of tool wear images via a milling-tool rotation and positioning algorithm based on Hough transform to facilitate boundary reconstruction and acquisition of wear assessment indicators; (4) determine the level of tool wear by reconstructing the boundary of the wear region of the tool tip through scanning feature points for positioning and least-squares fitting of lines. Finally, a tool wear in situ detection experiment was performed to analyze and verify the reliability of the tool wear algorithm proposed in this study and effectiveness of the fabricated detection system, demonstrating the tendency of tool wear indicators and sensitivity to the level of tool wear. The block diagram of the proposed research to be carried out in this paper is shown in Figure 1.

## 2. Materials and Methods

### 2.1. Machining Parameters

This paper aims to perform in situ tool wear detection during micro milling of a titanium alloy. For this purpose, Ti6Al4V was selected in this experiment, and its chemical composition is shown in Table 1. The thermal conductivity of titanium alloys is relatively low, only 1/7 of steel and 1/6 of aluminum. Thus, the heat produced when machining titanium alloy will not be transferred to the workpiece or taken away by chips, but gathered in the cutting area, and the generated temperature can be as high as 1000 °C or above [13,14], which rapidly causes the cutting edge of the tool to wear, crack and generate chip tumor, and a worn cutting edge appears, which causes the cutting area to produce more heat, further shortening the service life of the tool. To clearly represent wear behavior of the cutting tool during micro milling of the Ti6Al4V, dry cutting was employed. The cutting tool used in this experiment was a micro end milling cutter with a diameter of 1 mm, 2 flute, TiN coated, carbide end mill (JJ tools, Seoul, Korea), with the solid structure of the tool shown in Figure 2c. The specific geometry of the selected tool is shown in Figure 2d, where the clamping diameter (d) of the tool is 6 mm, the length of the cutting edge (L1) is 15 mm, the total length (L) is 55 mm, the taper angle of the connection part is 15°, and the nominal diameter of the cutting part (D) is 1 mm. In this work, two milling cutters were selected and numbered as Tn = 1, 2 respectively. In order to be able to observe the wear process and pattern of the microfine face milling cutter on the full cutting edge, the radial depth of cut should be equal to the tool radius value, and at the same time, a smaller axial depth of cut, spindle speed and feed rate should be selected to ensure that the micro end milling tool does not experience rapid wear in a short period of time. Therefore, the experimental parameters shown in Table 2 were developed by combining the parameter selection conducted in the pre-experiment.

Considering image processing algorithm and the verification procedure, we divided the micro milling experiment into two groups by the serial number of the cutting tools, and the cutting time in each group of the experiment was different. Micro end milling experiment No. 1 using Tn = 1 tool based on relevant parameters was conducted. The tool wear image gathered in the experiment was employed to verify the effectiveness of the image processing algorithm and complete the debugging of relevant image processing program. On the other hand, to understand the relationship between tool wear evaluation indicators and cutting time, micro end milling experiment No. 2 using Tn = 2 tool based on relevant parameters was performed and tool wear image was captured.

In this work, a VF-2SS Haas vertical machining center was used to conduct a micro end milling operation for in situ tool wear detection. Figure 2a illustrates the in situ tool wear detection system. The hardware of the tool wear image-gathering system used in the experiment is shown in Figure 2b, including (1) a telecentric lens; (2) a ring illuminator; (3) an industrial camera; (4) adapter plates; (5) a rotary platform; (6) Z axial support. The parameters of the industrial camera and the telecentric lens are shown in Table 3. The adapter plates and Z axial support in this system are designed to facilitate rapid connection with the machine tool and the rotary platform can be used to adequately adjust the camera, ensuring an accurate detection by placing the axis of the telecentric lens in parallel with that of the cutting tool. 

Micro end milling experiments were conducted based on parameters indicated in Table 2. Upon completion of every experiment, each milling cutter was purged by compressed air to remove chips. Then, the clean cutting tool was moved to the image inspection system to capture the tool wear in image form. Finally, tool wear was inspected and evaluated based on the generated image using an image processing program.

### 2.2. Types of Wear of Micro End Milling Cutter and Evaluation Indicator Selection

In the cutting process, severe friction occurs at tool-chip-workpiece interface, and heat and pressure are generated at such contact areas, leading to tool wear [15]. Regarding a turning tool, tool wear typically occurs on the rake face of the tool, which is also called crater wear, and the tool flank [16]. In addition, boundary wear is believed to be one of the types of tool wear, according to literature [15,17]. The wear mechanism involved mainly includes abrasive wear, adhesive wear, diffusion wear and oxidation (chemical) wear [16,17].

Each tooth of the conventional size end mill can be regarded as a turning tool [18], so its wear pattern is similar to that of a turning tool [16], and the main wear pattern is also front face wear and rear face wear, while the wear of the conventional size end mill is more severe at the rear face due to the more intense extrusion and sliding between the rear face of the tooth and the workpiece during milling [16], as shown in Figure 2.

Flank wear shown in Figure 3 represents all wear forms that may take place at a flank surface, including uniform flank wear (regular wear along the flank wear land), non-uniform flank wear (irregular wear along the flank wear land) and localized flank wear (wear often observed in any specific points within the flank wear land) [19]. In this figure, VB means the width of uniform flank wear, and *VBmax* is the maximum width of localized flank wear. The wear of the rake face is crater wear, and KT refers to the depth of the crater. In conventional size milling operations, *VBmax* and KT are typically selected for evaluating tool wear. Meanwhile, the wear of the original cutting edge is always ignored in the actual measurement of the wear of conventionally size milling tools due to such wear being relatively insignificant compared to wear band, and in most cases, only wear band is measured.

Micro milling is one of the common micro cutting methods, and tools used in the machining operation are relatively small. The range of the diameter of such cutting tools has not been definitive at present, but milling tools having a typical diameter between 0.1 mm and 1 mm are widely used [1,2,3,20]. Generally, milling tools of this size are 2-fluted cutters. Compared to conventional-size milling, micro milling exhibits a different machining mechanism due to some physical and mechanical properties, such as size effect and limit cutting thickness, that have insignificant influence during conventional milling play an important role in micro milling operations [21,22,23]. Thus, tool wear patterns between these two types of milling are not identical. As discussed, for conventional size milling operations, flank wear usually occurs. In contrast, tool wear is often observed at the tip of a tool in the micro milling process, which is also called tool tip wear. Specifically, the material at the tip of the tool is removed, and wear at the tip of the secondary flank surface is significant. Sometimes the coating of coated tools is found to peel off. Figure 4 indicates the types of wear of micro end milling tools.

Cutting tools need to be replaced if tool wear goes beyond a certain limit, which is called the tool-life criterion [17]. As most of tools witness flank wear and direct measurement of wear can be achieved with less effort, the width of flank wear rand (*VB*) identified at the 1/2 depth of cut can be used to formulate conventional size turning tool-life criteria as criterion ISO 3685. Unlike conventional size turning tools, tool wear of conventionally sized milling tools typically occurs at flank faces, so the width of flank wear rand (*VB*) is the most used according to criterion GB/T 16460-2016 [19]. Although there is no definitive criterion for evaluating the tool life of micro end milling tools, the following principle can be considered when choosing tool wear evaluation indicators according to the above two conventional size tools: (1) target types of wear must occur on tools; (2) evaluation indicators fit the wear process curve; (3) evaluation indicators are dominant and easily measured. It is known that tool wear usually occurs at the tip of micro end milling tools, especially the tip of secondary flank faces. Such wear is irregular wear in a subtriangular form. Furthermore, the helical cutter teeth of the main flank surface make it difficult to carry out in situ tool wear detection based on machine vision, facing a series of challenges such as positioning failure in circumferential direction, high magnitude noise during imaging of the helical surface and severe image segmentation error due to slight wear. Therefore, the maximum wear width of the secondary flank surface of micro end milling tools (*VBmax*) can be used to evaluate tool wear during micro end milling instead of the main flank face, considering the structural characteristics of tools. Aiming to evaluate tool wear in an accurate and comprehensive manner, this paper selects two more indicators, wear area *A_W_* at the tip of secondary flank face of cutters and reduction *Tdec* in diameter of bottom edge, to evaluate tool wear behavior in the micro end milling operations.

### 2.3. Tool Wear In-Situ Detection Image Processing Algorithm

To perform micro end milling experiments, two micro end milling tools were selected and named No. 1 and No. 2. Their original images (at the cutting time of 0 min) were gathered, as shown in Figure 5. 

To obtain the template image for image processing algorithm, the first micro milling experiment was conducted using tool No. 1 based on parameters shown in Table 2. Upon completion of the experiment, the tool wear image of tool No. 1 was generated. Figure 6 depicts the template image. It is seen that the wear form of No. 1 micro end-milling tool shown in the figure was consistent with the type of tool wear stated previously. Specifically, No. 1’s tool wear occurred mainly at the tip of two cutting teeth, and tool wear regions could be easily observed at its secondary flank surface, corresponding to wear region 1 and wear region 2, which are also the target zones to measure the amount of wear.

Figure 7 shows the flow chart of wear image processing algorithm of the micro end milling tool, shown as Figure 6, and the effect of each step, which will be discussed later in this section.

#### 2.3.1. Image Denoising (Image Pretreatment)

For machine vision-based detection, introducing noise into images during image transmission is inevitable and the noise may have a great impact on subsequent image processing. Therefore, noise generally is removed from images before segmenting the tool wear image and extracting features of tool wear, ensuring an accurate and reliable image-processing algorithm.

Noise in digital images mainly comes from two sources [24]:(1)image acquisition

When capturing images using CCD or CMOS technologies, noise is usually introduced when the sensor’s imaging performance is reduced due to adverse environmental conditions and sensor components such as light environment and sensor temperature. The noise encountered in such a case is Gaussian.

(2)image transmission

The interference of transmission channel gives rise to noise during image transmission. For example, noise, mainly salt-and-pepper noise, is easily introduced into images under atmospheric environment when transferring images via transmission medium wireless network.

Figure 6 shows that the image captured by the monitoring system exhibits high quality, and no visible noise points can be found. Therefore, the spatial filtering method can be used to remove the noise from the image. In contrast, the idea of the median filtering method is to sort all the gray areas of each pixel within the non-linear filter region into an order and then replace the pixel being calculated with the middle pixel value in a neighborhood of the pixels, reducing the level of intensity variation between one pixel and the next. Compared with the average filtering technique, median filtering is capable of locking image boundary and details. In this work, therefore, the median filtering technique was employed for image pretreatment to remove noise in images.

#### 2.3.2. Tool Wear Image Segmentation Based on Region Growing Method

Image segmentation is an integral part of the image processing task and the crucial foundation on which all image processing steps rely. The task of image segmentation is to separate the region of interest from the background for further analysis. At present, there is no all-purpose segmentation algorithm that can be used to achieve all image segmentation scenarios, so the segmentation performance of every algorithm needs testing to verify first under different circumstances. In this paper, the region growing method was used to segment tool wear images because it is accurate and efficient compared to other segmentation algorithms. Steps are as follows:(1)Assume that the original tool wear image is f(x,y), and threshold is *S*. The tool wear image was segmented to obtain a seed image S(x,y).
(1)$$S\left(x,y\right)=\{\begin{array}{c} 1,f\left(x,y\right)\geq S\\ 0,f\left(x,y\right)<S \end{array}$$(2)To obtain a new seed image, pixels in f(x,y) that meet the following conditions were added to the seed region S(x,y): (a) 8-connected neighborhood between pixels and seed points; (b) the absolute value of the gray level difference of pixels at f(x,y) and those in the original image corresponded by each seed point is less than or equal to the gray threshold T.(3)The region was iteratively grown by repeating Step (2) until no pixels that meet (a) and (b) can be picked to merge into the image. Then, the final segmented image g(x,y) was obtained.

#### 2.3.3. Tool Rotation and Positioning Algorithm Based on Hough Transform

The cutting edge in the template image for verifying the algorithm in Figure 6 is in an approximately horizontal position. When capturing a tool wear image of in situ detection, however, the spindle of the machine tool may stop at any position, leading to an uncertain direction of cutting edge in the gathered image. Thus, it is difficult to reconstruct the boundary and extract wear assessment indicators in the subsequent steps. To solve this problem, the Hough transform method was used in this study to detect the direction of the cutting edge, which provides an opportunity to rotate the cutting edge to a desired position.

Hough transform is the most used method for detecting straight lines and sometimes curves such as circles, ellipses and parabolas, within a given image [24,25]. Generally, all the points belonging to a line or a curve in an original image are mapped into a single point in the Hough domain. In other words, all such points from a peak in Hough space, transforming straight line or curve detection in an original image are used to finding limit values in the Hough domain. For example, when detecting straight lines using the Hough transform method, in polar coordinates, certain points in (x,y) domain were mapped to the sine-like line in the Hough domain (θ,ρ).
(2)xcosθ+ysinθ=ρ

Then, all points on a line in (x,y) domain corresponded to all sine-like lines in (θ,ρ) space that passed through the intersection point. Figure 8 illustrates the principle of the Hough transform method when detecting straight lines. During the running of the algorithm, the (θ,ρ) domain was separated into accumulator cells in a reasonable manner, as shown in Figure 8. Then, every point in (x,y) domain was transformed into a sine-like line in Hough domain, points are marked with the same color as their corresponding hough transformed curves for clear representation, and if the unit gird passed through by such a line, its value is increased cumulatively by 1. Finally, after the resulting peak values were identified, there (θ,ρ) was the parameter of the straight line, to be detected.

Based on the theory of Hough transform, the algorithm of micro end milling tool rotation and positioning can be operated as follows:(1)Sobel edge detection was performed for the segmented image to obtain the tool boundary image.(2)Hough transform was used to detect the straight line of the cutting edge from the tool boundary image obtained in Step (1) to identify the angle of inclination of the cutting edge.(3)The micro end milling tool was rotated if necessary to a horizontal position to obtain a horizontal image of the cutting edge.

#### 2.3.4. Boundary Reconstruction of Wear Region of Tool Tip

After image segmentation and the cutting edge having been rotated to a desired position, the tool tip wear region needs to be reconstructed before extracting the tool wear region. Next, the wear region of the tip of tool No. 1’s tooth was reconstructed (No. 2’s tooth follows the same principle), as shown in Figure 9. The reconstruction algorithm in this study will be run as follows:
(1)For the horizontal image of the cutting edge, Canny edge detection was used to detect the tool’s contour and extract the boundary of the tool.(2)The detected image was scanned through, row by row, from top to bottom until the pixel point with a value of 1 was identified. The coordinate point of the pixel was recorded, and it was the top corner A of the tool. Similarly, the bottom corner B was identified by scanning through the image, row by row from bottom to top.(3)The midpoint of A and B was computed based on the features of the tool, which was the central point O of the secondary flank face of the tool.(4)The boundary image extracted by Canny edge detector was scanned through, column by column from left to right, until the pixel with a value of 1 was identified. The coordinate point of the pixel was recorded, and it was the left corner C of the tool. Similarly, the right corner D was identified by scanning through the image column by column from right to left.(5)Taking O of the tool as the coordinate reference point to define the range of the column coordinate for scanning area 1. Within this target region, scanning was performed column by column from left to right, and from bottom to top for each column to locate the coordinate of the pixel with a value of 1 until coordinates of all points in scanning area 1 were obtained.(6)Taking O and C of the tool as the coordinate reference point to identify the area of the row coordinate for scanning area 2. The defined area was scanned through, row by row from top to bottom, and from left to right for each row to identify the coordinate of the pixel with a value of 1 until the coordinates of all points in scanning area 2 were obtained.(7)All points in scanning area 1 obtained in Step (5) were fitted by least squares to get the equation of straight line $l_{1}$, and the equation of straight line $l_{2}$ was obtained by fitting all points in scanning area 2 identified in Step (6) to the first-order polynomial by least squares.

Fitting and reconstructing lines using the least-squares polynomial fitting method is capable of achieving subpixel-level positioning [26] and obtaining high precision detection [27]. The principle of the algorithm is to assume that the pixel set to be fitted is $\left(x_{1},y_{1}\right)$, $\left(x_{2},y_{2}\right)$∗$\left(x_{n},y_{n}\right)$, and the linear regression equation of the straight line can be expressed:(3)y=ax+b

Then a and b are parameters of the equation of lines to be fitted if the below equation is met.
(4)$$e=\sum_{i=1}^{n}{\left(ax_{i}+b-y_{i}\right)}^{2}$$

#### 2.3.5. Extraction of the Amount of Tool Wear

As discussed, the indicators identified to evaluate tool wear behavior during micro milling include maximum wear width of the secondary flank surface at the tip of a tool, wear area at the tip of secondary flank face of a tool and the reduction in diameter of the bottom edge. Next, detailed detection and computation of each indicator will be introduced.

Wear width and wear area

To detect the amount of tool wear, the wear region of the tip of tool No. 1’s tooth was selected, as shown in Figure 10; polygon CFG is the worn region. According to the polynomial equation of $l_{1}$ and $l_{2}$ obtained in Section 2.3.3, suppose that $l_{1}:y=a_{1}x+b_{1}$, $l_{2}:y=a_{2}x+b_{2}$. 

It is known that C indicates the left corner of the worn tool tip and has been involved in fitting of $l_{2}$, and hence, C is believed to be on or very close to the straight line $l_{2}$. Therefore, the length of CE, |CE| can be deemed to be the maximum wear width of the subtriangular CFE wear region. Then, it is assumed that any point on the wear boundary is $\left(x_{i},y_{i}\right)$, then the distance between $\left(x_{i},y_{i}\right)$ and the original cutting edge $l_{1}$ can be used as the wear width of the wear boundary point $VB_{i}$. Thus, the maximum width of all wear boundary points from C to G can be obtained, which is the maximum wear width $VB_{max}$. On the other hand, wear area *A_W_* can be determined as it consists of the CFE area and CEG area, among which, the CEG area can be obtained by computing the total number of pixels within CEG.

To measure the wear extent of the tip of tool No. 1’s tooth, the maximum wear width and wear area algorithm can be operated as follows:

(1) For the boundary image obtained by Canny edge detector during reconstructing the boundary of wear region of tool tip, scanning was conducted, column by column from left to right, by taking the column coordinate of C obtained by reconstruction algorithm as the starting column. Every column was scanned from bottom to top, and the first pixel with value of 1 identified was the wear boundary point $\left(x_{i},y_{i}\right)$, and the distance $VB_{i}$ from $\left(x_{i},y_{i}\right)$ to $l_{1}$ can be computed:(5)$$VB_{i}=\left|\frac{a_{1}x_{i}-y_{i}+b_{1}}{\sqrt{a_{1}^{2}+{\left(-1\right)}^{2}}}\right|$$

(2) If $VB_{i}$ in step (1) is less than 1 pixel, the scanning operation was stopped to obtain the wear boundary point $\left(x_{1},y_{1}\right)$, $\left(x_{2},y_{2}\right)$∗ $\left(x_{n},y_{n}\right)$, and then the maximum wear width was:(6)VBmax=V1≤i≤nmaxBi

(3) The coordinate of F at the tip of the tool $\left(x_{F},y_{F}\right)$ was obtained. F was the intersection point of $l_{1}$ and $l_{2}$, and meets the following system of equations:(7)$$\{\begin{array}{c} y_{F}=a_{1}x_{F}+b_{1}\\ y_{F}=a_{2}x_{F}+b_{2} \end{array}$$

As shown in figure, straight line CE passes through *C*
$\left(x_{C},y_{C}\right)$, and is perpendicular to $l_{1}$. Assume that the equation of the straight-line CE is $l_{\mathrm{CE}}:-x+a_{1}y+c_{3}=0$, then $c_{3}$ meets the below equation:(8)$$-x_{C}+a_{1}y_{C}+c_{3}=0$$

(4) Then wear area *A_W_* can be expressed as:(9)$$\left|CE\right|=\left|\frac{a_{1}x_{C}-y_{C}+b_{1}}{\sqrt{a_{1}^{2}+{\left(-1\right)}^{2}}}\right|$$
(10)$$\left|FE\right|=\left|\frac{a_{3}x_{F}-y_{F}+b_{3}}{\sqrt{a_{1}^{2}+{\left(-1\right)}^{2}}}\right|$$
(11)$$A_{W}=\frac{1}{2}\left|FE\right|\left|CE\right|+\sum_{i=1}^{n}VB_{i}$$

2.Reduction in diameter of bottom edge

Extracting the features of the diameter of the tool bottom edge can be achieved by the smallest enclosing circle of the worn tool boundary in the boundary image identified by Canny edge detector in the wear region boundary reconstruction algorithm. In this paper, a classic algorithm to find the smallest circle that completely contains a set of points proposed by Wang Wei was used, as it is accurate and efficient [28,29]. The algorithm was operated as follows:

(1) Three points $P_{1}$, $P_{2}$ and $P_{3}\;$ were randomly selected from the set of points in boundary image identified by Canny edge detector.

(2) A smallest enclosing circle C was drawn by $P_{1}$, $P_{2}$ and $P_{3}$.

(3) $P_{4}\;$ that is farthest from the center of the circle C in the set of points of the boundary image was found. The algorithm stopped if $P_{4}$ was on or within C.

(4) Three points were selected from $P_{1}$, $P_{2}$, $P_{3}$ and $P_{4}$ in a way to ensure a smallest enclosing circle can be constructed to contain the four points. Then the selected three points were new $P_{1}$, $P_{2}$ and $P_{3}$. Repeat Step (2).

The smallest enclosing circle algorithm makes it possible to determine the radius r of the minimum enclosing circle of the worn tool and the radius R of the smallest enclosing circle of a new tool; thereby, the reduction in diameter $T_{dec}$ can be expressed as:(12)$$T_{dec}=2R-2r$$

## 3. Experimental Result and Analysis

### 3.1. Actual Result Analysis for Tool Wear In-Situ Detection Algorithm

According to the flow chart of tool wear image processing algorithm, as shown in Figure 7, the median filtering method was used to remove noise from the template image shown in Figure 6.

The principle and procedures of the region growing method indicate that the critical steps of the method are seed pixel selection, growing rule and a stop condition as they have a significant impact on the final segmentation. Figure 11a is the tool wear image in which the noise was removed. In the figure, bright spots were observed on the secondary flank surface of the tool, and pixels with a maximum gray value of 255 were detected. Therefore, a seed pixel can be chosen based on S = 255, as shown in Figure 11c. In this study, gray-level difference was used as an indicator for measuring pixel similarity. Figure 11b indicates a gray level histogram of tool wear image. It is seen that the gray value was between 0~35, and the gray difference was 220. Thus, a threshold of T = 220 was selected. Figure 11d is the segmented image and exhibited good segmentation performance as compared to Figure 11a, obtaining an accurate tool boundary using the region growing method. Furthermore, it is seen from Figure 11d that tiny holes were observed in the binary image, so a morphological hole filling method was used to fill holes before the final segmented image was obtained [24].

Figure 12b shows the image treated by morphological hole filling method, and it exhibited good filling performance as compared to Figure 12a. It also can be seen in Figure 12c that this method did not change the features of the tool boundary, but filled the tiny holes, and hence it will not have any impact on subsequent tool wear extraction.

The boundary of the segmented image was detected, as shown in Figure 13b. Then, Hough transform was performed for the image, as indicated in Figure 13c. Figure 13d shows the two longest straight lines mapped in the segmented image. It is observed in Figure 13a that the areas (No. 1~No. 4) containing longer straight lines were identified on the worn tool after analyzing the segmented image. In Figure 13c, two maximum peaks were found in the Hough transform space of the tool wear image, i.e., the neighboring red blocks in the image. It is seen that θ of the two values were identical, indicating that the two longest and parallel lines are available at the enclosing boundary edge of the worn tool. Figure 13d illustrates that two longest straight lines obtained by Hough transform were still on the cutting edge, although the tool has been severely worn. This indirectly proves that secondary flank wear of micro end milling tools usually occurred on the tip of tools, while the wear region on the cutting edge was insignificant. 

To verify the effectiveness of tool rotation and positioning algorithm based on Hough transform, the segmented image shown in Figure 13a was rotated to simulate every possible angle of a cutting edge that may occur in tool wear image. The tool rotation and positioning algorithm discussed in the paper was adopted to test the images with different angles, as shown in Figure 14. It can be seen that tool rotation and positioning based on Hough transform exhibited good performance and reliability, and proved to be an effective method to accurately locate cutting edges with different angles.

Based on the boundary reconstruction algorithm discussed in Section 2.3.4, the boundary of the segmented and positioned tool wear image was reconstructed, as shown in Figure 14d. The results of each reconstruction step were plotted on the rotated original wear image, as shown in Figure 15. The boundary corner point of the tool A~D and the central point O, referring to the schematic diagram in Figure 9, are represented by red spots, the blue spots are the set of points identified to be fitted for reconstruction algorithm, and the green lines are fitted straight lines by the least squares method. The partial enlarged drawing shows that the reconstruction algorithm proposed in the paper is an effective and accurate method to achieve reconstruction of the wear region of tool tip. 

Analysis in Section 2.3.5 shows that the accuracy of wear amount extraction algorithm depends on the extraction of wear boundary and straight-line CE. To verify the effectiveness of the algorithm, the wear boundary points, and straight-line CE were shown in rotated original wear image. In Figure 16, yellow spots are boundary points identified by the wear extent extraction algorithm, the red line is straight-line CE, the red points are the boundary vertices of the wear area CFE and CEG, refer to the schematic diagram in Figure 10 for specific details. The partial enlarged drawing indicates that the wear extent extraction algorithm proposed in this study is an effective method to extract the wear amount of a worn tool. 

To run an algorithm of reduction in diameter of bottom edge, the first step was to identify Canny boundary of the tool image, and then the smallest enclosing circle of the Canny boundary was found. Aiming to present the matching accuracy of the smallest enclosing circle, it was indicated on the Canny boundary of the tool wear template image, as shown in Figure 17. Figure 17a shows that the smallest enclosing circle of the Canny boundary obtained by the smallest enclosing circle algorithm accurately fit to every pixel on the Canny boundary, especially the outermost point, ensuring an accurate diameter value at bottom edge based on the smallest enclosing circle.

### 3.2. Tool Wear In-Situ Detection Result and Change Tendency Analysis during Micro End Milling

Tool No. 2 was used to conduct a micro milling operation to understand the change tendencies of wear evaluation indicators. Figure 18 shows tool wear in situ detection images at different cutting times. It can be seen that in all four sub-pictures of Figure 18, speckled areas appear and their area gradually increases step by step as the cutting progresses. Since microfine face milling cutters are used for milling titanium alloys, the form of tool wear is dominated by abrasive wear and adhesive wear at lower cutting speeds [30]. Therefore, the speckled area in Figure 18 shows that the main form of wear on the sub-rear face of the micro face milling cutter is adhesive wear. At the same time, the tip area of the micro face mill is subjected to greater cutting thermal stresses as it is involved in the radial and axial cutting of the workpiece material. Furthermore, because of the small size of the tip area of the micro end mill and the large contact area, it will cause the tip area to break earlier, which is the reason for the change of the tip area in Figure 18. At the same time, there are obvious signs of diffuse wear along the radial direction on the cutting edge of the tool tooth 1 in Figure 18b.

It is seen that tool wear mainly occurred at the tip of the tool tooth, and the wear extent of the secondary flank surfaces of tool tooth 1 and tool tooth 2 was uneven due to radial runout error of the spindle and tool installation error. It can be found that the wear amount at the tip of tool tooth 1 was significantly greater than that of tool tooth 2. Within the specified cutting time in the micro milling experiment, tool tooth 1 suffered more severe wear than tool tooth 2 and witnessed a great loss of tool material from the tool tip. Therefore, the algorithm was only applied to indicators of wear amount for tool tooth 1, and the detection results are shown in Table 4. 

In Table 4, the unit of the maximum wear width $VB_{max}$ and reduction in diameter $T_{dec}$ is the actual length of a single pixel, and the unit of wear area *A_W_* is equal to the actual area of a single pixel. Figure 19 shows the change tendency chart of every wear evaluation indicator.

The change tendency images of every wear evaluation indicator shows that an increase in cutting time caused the maximum wear width $VB_{max}$, wear area *A_W_* and reduction in diameter $T_{dec}$ to increase linearly and wear amount to increase. By comparing the increments and change tendencies of every evaluation indicator under different cutting times in Figure 19, it can be seen that the growth rate of the maximum wear width $VB_{max}$ wear area *A_W_* and reduction in diameter $T_{dec}$, at the starting stage (Ct = 1 min), were much higher than those at the subsequent stage (Ct = 4, 7, 10 min) where the relative increment and rate of change became stable. This is in line with the conclusion reached in the previous analysis of Figure 18, that the tip of the micro end mill is more prone to wear at the beginning of tool wear. In comparison, the trend of the relative growth of the wear index is different, with a certain fluctuation for the wear width $VB_{max}$ and diameter reduction $T_{dec}$, while the relative growth of the wear area *A_W_* decreases gradually with the increase of the cutting time. This shows that the main factor for the increase in wear area *A_W_* is the wear at the tip of the tool, whereas the leading cause of the increase in wear width $VB_{max}\;$ and diameter reduction $T_{dec}$ is wear at the tip, there is also diffuse wear along the cutting edge in the radial direction, and the effect of such wear gradually increases as the cutting progresses. Compared to the maximum wear width $VB_{max}$ and reduction in diameter $T_{dec}$, the wear area *A_W_* witnessed a wide range of values, which helps to present wear behavior more efficiently. This is because wear area is a two-dimensional measurement indicator to identify both wear width and length of a wear region.

The above conclusion on the wear pattern of micro end milling tools proves the correctness of the aforementioned analysis of the wear quantity characterization indicators; the selection of indicators is discussed in Section 2.2 above. As a matter of fact, more wear indicators can be monitored in practical applications to understand tool wear behavior in a comprehensive manner, as reliability may not be ensured by detecting only a single indicator. The principle of selecting the corresponding wear evaluation index is to use the wear area *A_W_* as the main judgment of the wear state at the initial stage, and the wear width $VB_{max}$ and diameter reduction $T_{dec}$ can be used as the main inspection index to judge the wear state of the tool as the cutting progresses.

## 4. Conclusions

To achieve the in situ detection of the wear status of micro end milling tools, the main research conducted in this paper and the results obtained are as follows:(1)This paper firstly builds a micro milling tool wear image acquisition system based on machine vision design according to the in situ detection requirements, and then analyzes the wear morphological characteristics of micro end milling tools. By comparing the wear pattern with that of conventional size end mills, the corresponding evaluation indictors of the wear state of micro face milling tool are proposed as (1) wear area *A_W_*, (2) maximum width of wear $VB_{max}$, (3) diameter reduction *Tdec*.(2)Meanwhile, the corresponding image processing process and corresponding algorithm algorithms are proposed for the processing of the acquired tool wear template images, which are used to realize: 1. source image denoising; 2. segmentation and extraction of the tool contour and envelope region; 3. rotational positioning of the tool; 4. edge reconstruction of the tool wear region; 5. extraction and calculation of the wear amount, respectively. The detection function of the tool wear status evaluation indictors is realized by writing the corresponding computer program.(3)Then, the accuracy and reliability of the proposed algorithm and the preparation procedure were verified through the analysis of the results of the verification tool wear images, and the variation law of the wear amount of the micro end milling tool with the cutting time was obtained. By analyzing the law, it can be seen that the wear area *A_W_* is more effective for detecting the wear at the tool tip in the first cutting stage, and the wear width $VB_{max}$ and diameter reduction *Tdec* can be used for detecting tool wear in the subsequent cutting stage.(4)Finally, in order to improve the accuracy when performing in situ detection of tool wear on micro end mills, two aspects can be investigated in future work: (1) Improving the relevant image processing algorithms to improve the accuracy of image detection; (2) Increasing the types of acquired signals and using artificial intelligence algorithms to establish the correspondence between images and the remaining signals to make the in situ monitoring of tool wear status more accurate.

## Figures and Tables

**Figure 1 micromachines-14-00100-f001:**
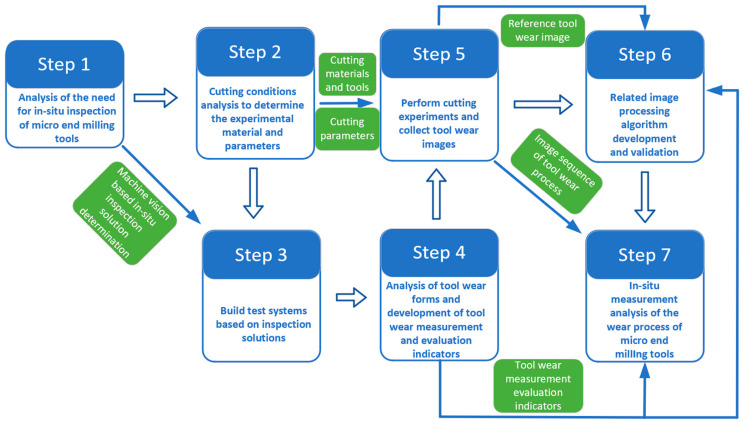
The block diagram of the proposed research.

**Figure 2 micromachines-14-00100-f002:**
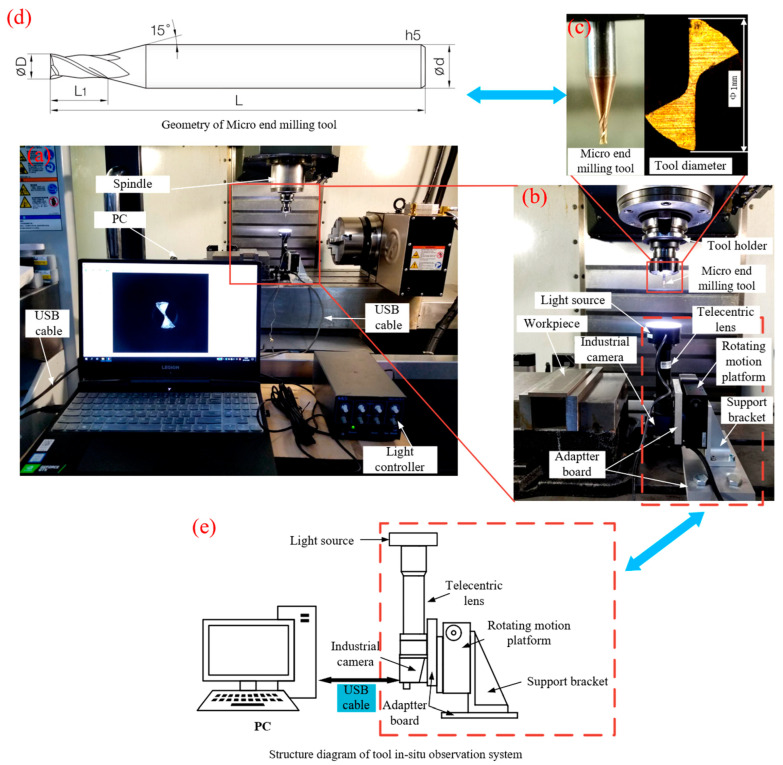
Pictures of in-situ tool wear detection system of micro milling tool:(**a**) Overview of in situ tool wear detection system of micro milling tool; (**b**) Hardware of image-gathering system of micro milling tool; (**c**) micro end milling tool; (**d**) Geometry of micro end milling tool; (**e**) Structure diagram of tool in situ observation system.

**Figure 3 micromachines-14-00100-f003:**
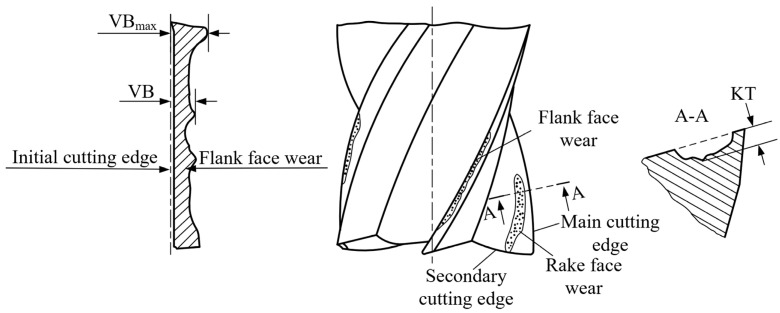
Wear Behavior of a Conventional Size End Milling Tool.

**Figure 4 micromachines-14-00100-f004:**
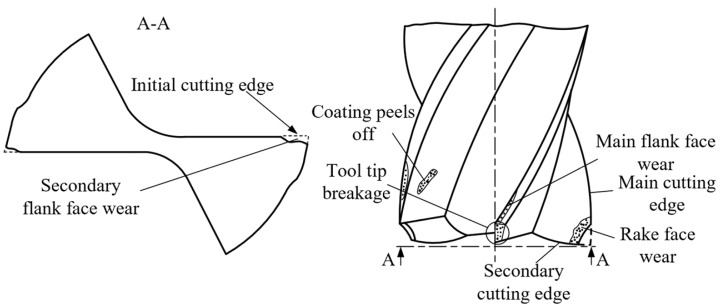
Types of Wear of Micro End Milling Tools.

**Figure 5 micromachines-14-00100-f005:**
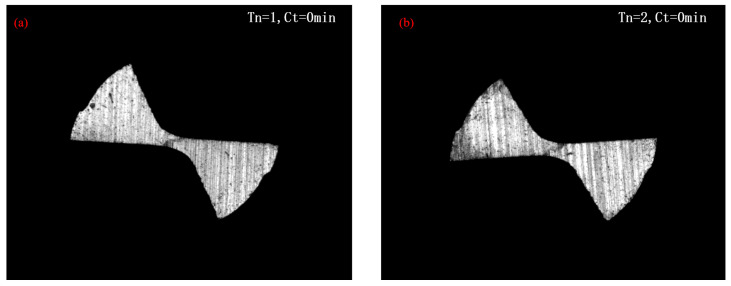
Original images of micro end milling tools: (**a**) original image of No. 1 tool; (**b**) original image of No. 2 tool.

**Figure 6 micromachines-14-00100-f006:**
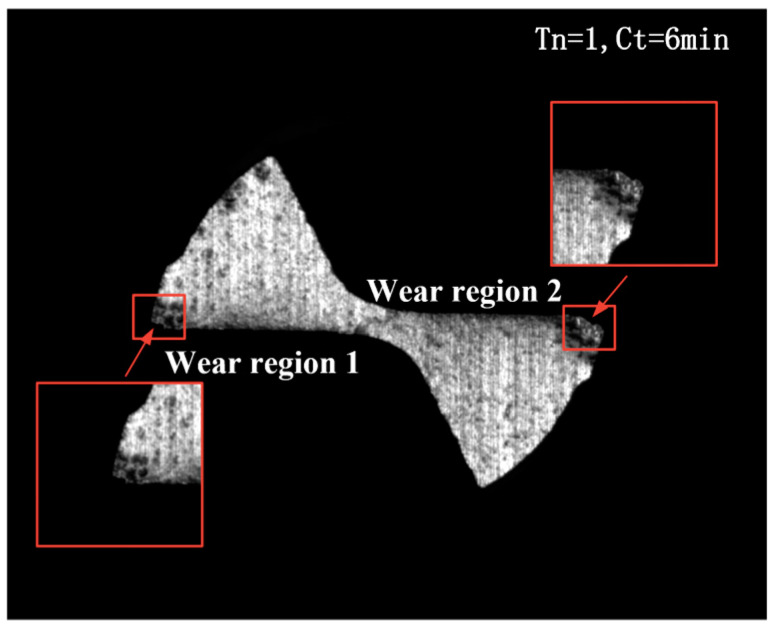
Template image developed for image processing algorithm of tool wear in situ monitoring of micro end milling tool cutter.

**Figure 7 micromachines-14-00100-f007:**
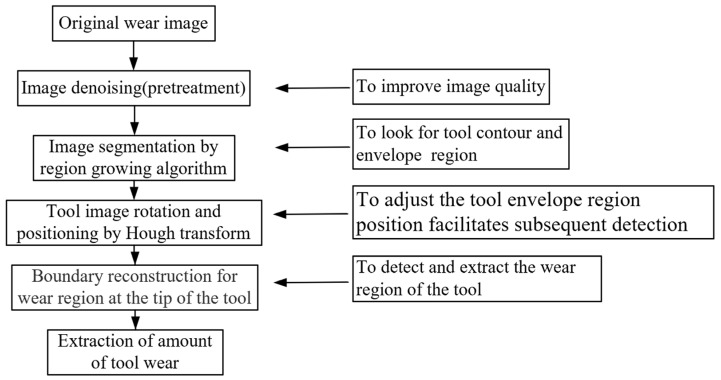
Flow chart of tool wear image processing algorithm of micro end milling cutter.

**Figure 8 micromachines-14-00100-f008:**
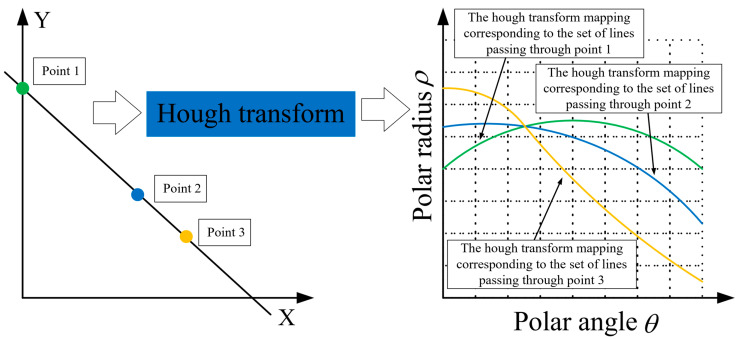
Principle of Hough transform in detection of straight lines.

**Figure 9 micromachines-14-00100-f009:**
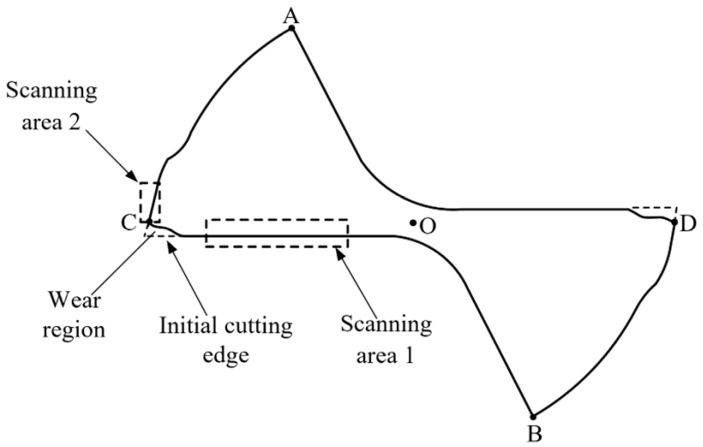
Schematic of tool wear region boundary reconstruction.

**Figure 10 micromachines-14-00100-f010:**
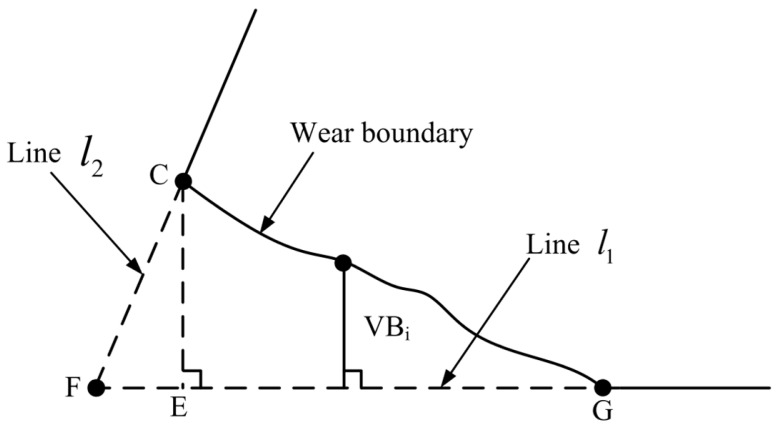
Schematic of tool wear extent extraction.

**Figure 11 micromachines-14-00100-f011:**
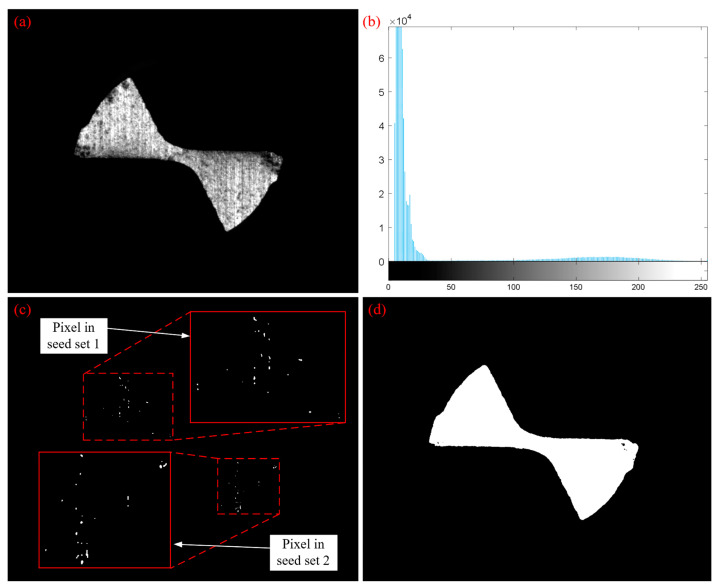
Tool wear image segmentation based on region growing method: (**a**) denoised tool wear image; (**b**) a gray level histogram of (**a**); (**c**) seed pixel selection image; (**d**) segmented image.

**Figure 12 micromachines-14-00100-f012:**
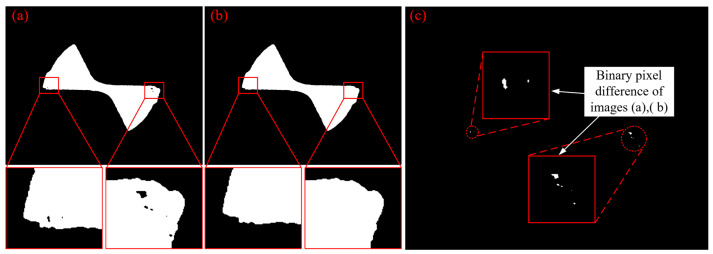
Morphological hole filling result: (**a**) segmented image by region growing method; (**b**) image after morphological hole filling; (**c**) comparison image before and after filling holes.

**Figure 13 micromachines-14-00100-f013:**
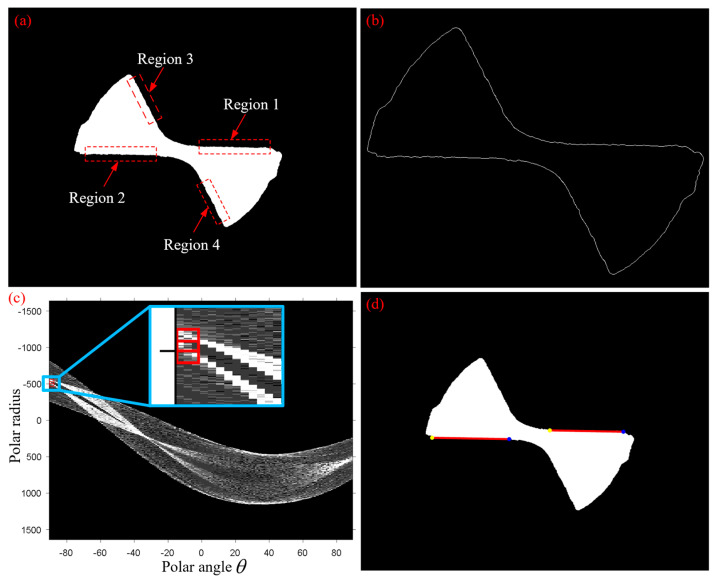
Hough transform of tool wear image: (**a**) segmented image; (**b**) boundary obtained by Sobel edge detector; (**c**) Hough transform of (**b**); (**d**) two longest straight lines in the original image.

**Figure 14 micromachines-14-00100-f014:**
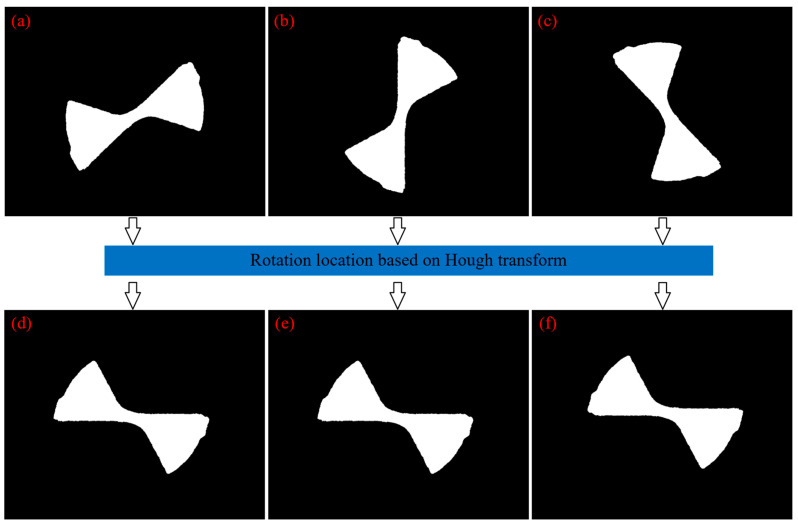
Simulation of tool rotation and positioning based on Hough transform: (**a**) 0~90°; (**b**) 90°; (**c**) 90~180°; (**d**–**f**) Images after rotation and positioning.

**Figure 15 micromachines-14-00100-f015:**
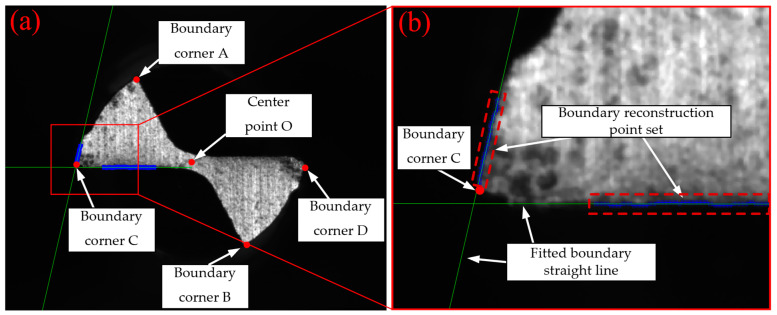
Tool tip wear region boundary reconstruction in rotated original wear image: (**a**) overall image of the wear region boundary reconstruction; (**b**) detail image of the wear region boundary reconstruction.

**Figure 16 micromachines-14-00100-f016:**
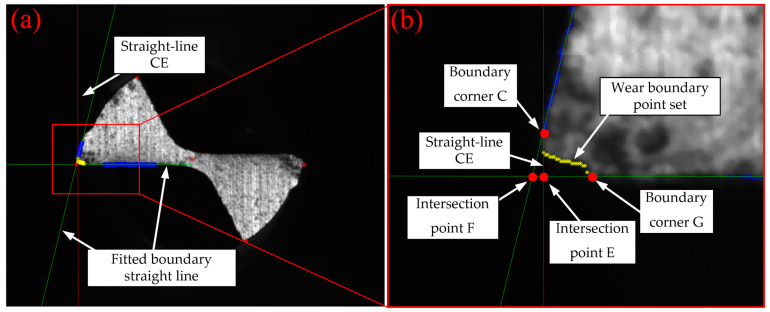
Wear extent extraction in rotated original tool wear image: (**a**) overall image for the wear extent extraction; (**b**) local detail image for the wear extent extraction.

**Figure 17 micromachines-14-00100-f017:**
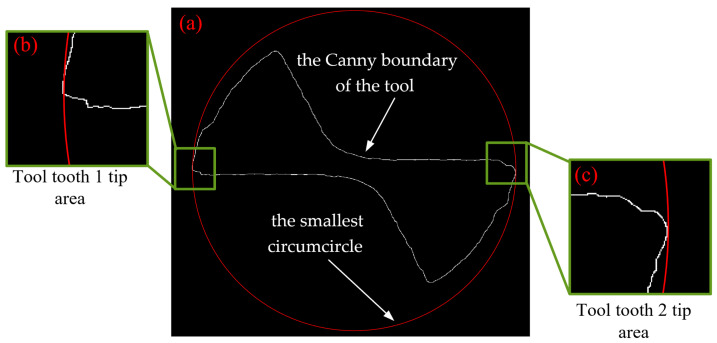
Smallest enclosing circle algorithm in the Canny boundary image: (**a**) boundary extracted by Canny edge detector with the smallest circumcircle shown in the Canny boundary image; (**b**) the detail of tool tooth 1 tip area; (**c**) the detail of tool tooth 2 tip area.

**Figure 18 micromachines-14-00100-f018:**
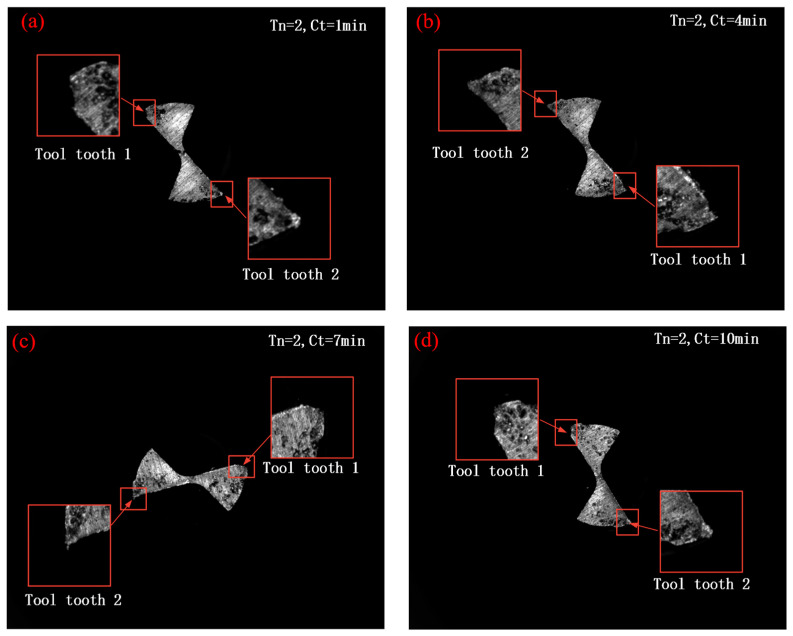
In situ captured wear images at different cutting times of tool No. 2: (**a**) cutting time 1 min; (**b**) cutting time 4 min; (**c**) cutting time 7 min; (**d**) cutting time 10 min.

**Figure 19 micromachines-14-00100-f019:**
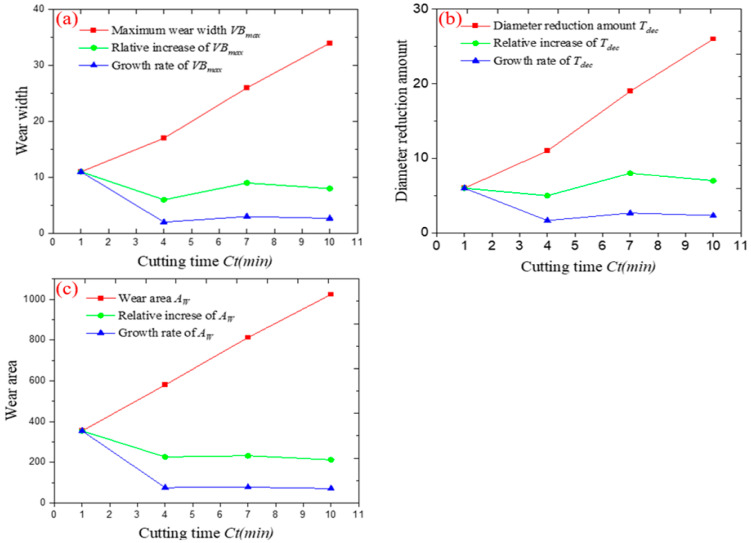
Change tendency chart of wear evaluation indicator of tool No. 2: (**a**) change tendency chart of maximum wear width $VB_{max}$; (**b**) change tendency chart of diameter reduction $T_{dec}$; (**c**) change tendency chart of wear area *A_W_*.

**Table 1 micromachines-14-00100-t001:** Chemical Composition of Ti6Al4V.

Chemical Composition	Al	V	Fe	Si	O	C	N	Others
Specific weight	5.5~6.5	3.5~4.5	0.25	0.15	0.13	0.08	0.05	0.512

**Table 2 micromachines-14-00100-t002:** Machining parameters used during micro end milling.

Machine Tool	Tool No. (Tn)	Spindle Speed (S r/min)	Axial Depth of Cut (ar mm)	Radial Depth of Cut (ae mm)	Feed Speed (F mm/min)	Cutting Time (Ct min)
Haas vertical machining center (VF-2SS)	1	7000	0.02	0.500	500	6
	2	7000	0.02	0.500	500	1,4,7,10

**Table 3 micromachines-14-00100-t003:** Parameters of industrial camera and telecentric lens.

Telecentric Lens Parameters		Industrial Camera Parameters	
type	non-coaxial	resolution	2448 × 2048
magnification	3	sensor	2/3-inch CMOS
type of lens mount	C	output interface	USB3.0
distortion	<0.1%	frame rate	79 fps
maximum compatible size of a camera	2/3 inch	pixel size	3.45 μm
depth of focus	0.5 mm	type of lens mount	C
working distance	65 mm	color	black and white

**Table 4 micromachines-14-00100-t004:** In situ wear detection result of No. 2 tool (unit: length/area of a single pixel).

No. 2 Tool	Cutting Time	The Maximum Wear Width $VB_{max}$	Wear Area *A_W_*	Reduction in Diameter $T_{dec}$
2 flute, TiN-coated carbide end mills with a diameter of 1 mm	1 min	11	354	6
	4 min	17	580	11
	7 min	26	812	19
	10 min	34	1024	26

## Data Availability

Experimental data can be obtained by requesting from X.Z. or Z.Y.

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
