# Peer review of "Study on In-Situ Tool Wear Detection during Micro End Milling Based on Machine Vision"

_micromachines, 2022, doi:10.3390/mi14010100_

Round 1
Reviewer 1 Report
Comments:
1.Section 2.1: The temperature of 1000 0C should be referenced.
2. The geometry of the end mill should be presented
3. How did authors select the values of process parameters.
4. The discussion on the tool wear seems very superficial.
5. The conclusions should be concise and to the points.
6. The physical understanding for Fig. 18 is hard to find in the article.
7. Authors may enhance the introduction by adding some recently published articles such as, https://doi.org/10.1016/j.jmapro.2022.07.053, https://doi.org/10.1115/1.4053315, https://doi.org/10.1016/j.matlet.2022.133078,
Author Response
Point 1: Section 2.1: The temperature of 1000 should be referenced.
Response 1:
Based on the reviewer's comments, the relevant citation was added to the discussion of cutting temperature when cutting titanium alloys in section 2.1: The temperature of 1000 ℃, and the corresponding references 13 and 14 were added to the references.
Point 2: The geometry of the end mill should be presented.
Response 2:
According to the reviewer's comments, the geometry of the micro milling cutter was added to Figure 2, i.e., Figure 2-(d), and the introduction and values of the relevant structural parameters were added to the text; the cutting parts of the tool geometry in Figure 3 and Figure 4 were marked with the corresponding cutting edges.
Point 3: How did authors select the values of process parameters.
Response 3:
In order to be able to completely observe the wear form of the selected micro end milling tool on the bottom edge, the radial depth of cut should be greater than or equal to 50% of the radius of the tool, and the cutting method in this paper is dry cutting, such as the use of too high spindle speed, too large axial depth of cut and too high feed rate will cause the micro end milling tool sharply worn or even burn breakage in a short period of time, it is not easy to observe the normal wear form and law of the micro end milling tool. Therefore, based on the selected tool diameter of 1mm, the radial depth of cut is set to 0.5mm in this paper to ensure that the entire bottom edge of the tool can be involved in cutting during micro milling. At the same time, by selecting a smaller spindle speed, axial depth of cut and feed rate, the sharp wear of the tool in a short period of time can be delayed and the in-situ observation of the normal wear form and pattern of the micro end milling tool can be realized. In this case, the selection of cutting time was developed in conjunction with the statistical law of effective tool life obtained when milling titanium alloys with this type of tool. Finally, based on the reviewers' comments, the reasons for the selection of cutting parameters are described in the paper.
Point 4: The discussion on the tool wear seems very superficial.
Response 4:
Based on the reviewer's comments, firstly, the four inspection images of the sub-rear tool face of the No. 2 tool at different cutting times in Fig. 18 were organized, and the markings about the tool teeth were added, and the details were enlarged so that the changing forms of tool wear could be further reflected clearly. Based on this basis, the wear forms of each tooth of the No. 2 tool at different cutting times were analyzed, and the explanation of the mechanism corresponding to the tool wear forms was added. The statistical graphs of tool wear of No. 2 in Fig. 19 and Table 4 were reanalyzed and understood, and the trends of wear indicators were explained at the level of wear mechanism based on the wear mechanism of micro-end milling tools obtained in the pre-experiments.
Point 5: The conclusions should be concise and to the points.
Response 5:
The conclusion of the paper was reorganized according to the reviewer's comments; at the same time, the work performed and the conclusions obtained were summarized as points in the paper for clarity, and the introduction of future research work was added, corresponding to point 4 in the conclusion of the paper.
Point 6: The physical understanding for Fig. 18 is hard to find in the article.
Response 6:
Due to the addition of a block diagram showing the contents of the proposed research in the beginning part of the paper, the numbering of Figure 18 in the paper is extended to Figure 19. Based on the reviewer's comments, the trend of the relevant wear indicators in Figure 19 is rediscussed and the main influencing factors of their corresponding changes are analyzed. On the basis of this discussion the validity of the tool wear evaluation selected in the previous subsection 2.2 can be verified, and the effective principle of using the tool wear evaluation indicators in a reasonable way for the different stages of the micro end milling in practical applications is given.
Point 7: Authors may enhance the introduction by adding some recently published articles such as, https://doi.org/10.1016/j.jmapro.2022.07.053, https://doi.org/10.1115/1.4053315, https://doi.org/10.1016/j.matlet.2022.133078.
Response 7:
The recent references were added according to the reviewer's opinion, corresponding to the references labeled [4], [5], [6], [7], [11] and [12], Among them, references [4], [5], [6], and [7] introduce the current research status of the indirect tool state prediction method, and its new progress is implemented in the signal feature recognition algorithm with the help of machine learning; references [11] and [12] are studied in the area of tool wear detection based on machine vision, with the same newer advances introducing machine learning methods for the recognition of wear in tool images, but mostly for features for recognition, although mostly for conventional size milling tools.
Please also see the attachment.

Reviewer 2 Report
-The paper is interesting ;;;
-it is a good idea to add a block diagram of the proposed research (step by step);;;
-it is a good idea to add more photos of measurements, sensors + arrows/labels what is what (if any);;;
-What is the result of the analysis?;;
-figures should have high quality. ;;;;;
-text should be formatted;;;;
-Is there a possibility to use the proposed research for other topics, classification methods etc.:
"Thermographic Fault Diagnosis of Ventilation in BLDC Motors";;;;
-please compare advantages/disadvantages of other approaches etc.;;;
-please add some sentences about future work;;;
Author Response
Point 1: It is a good idea to add a block diagram of the proposed research (step by step).
Response 1:
As suggested by the reviewer, a block diagram was added to the paper to show the content of the proposed research to be carried out, as shown in Figure 1 added to the paper.
Point 2: It is a good idea to add more photos of measurements, sensors + arrows/labels what is what (if any).
Response 2:
Based on the reviewer's suggestion, a schematic diagram of the structure of the in-situ tool wear measurement system has been added, as shown in Figure 2-e; at the same time, corresponding annotation labels have been added to the relevant components of the in-situ detection system in the figure.
Point 3: What is the result of the analysis?
Response 3:
Based on the reviewer's comments, the conclusion part of the analysis was reorganized in this paper, and the revised result analysis consists of two main parts: 1. The proposed image processing algorithm is examined using the wear image of tool No. 1 as a template image to verify the effectiveness and accuracy of the proposed algorithm for in-situ wear detection of micro end milling tools. 2. In-situ detection of the wear status of tool No. 2 is performed, and the the analysis of the measurement results verifies the correctness and sensitivity of the wear evaluation indicators of the micro end milling tool proposed in Subsection 2.2 above. At the same time, the causes of the main forms of tool wear were analyzed mechanistically according to the acquired tool wear image sequences. Finally, the evolution of the corresponding wear evaluation indicators is obtained from the statistics of tool wear, and on this basis, a reasonable selection principle of the tool wear evaluation indicators in the actual inspection is obtained.
Point 4: Figures should have high quality.
Response 4:
Based on the reviewer's comments, the contrast of the images in the paper was increased to improve the image quality, and the corresponding detail display was added to the result images to make them clearer and more distinct. The work carried out is: 1. To address the unclear display of the growth seeds set in Figure 10-(c) when the image is segmented, the corresponding local details are added to the display figure; 2. To reflect the effect of the image ecology algorithm in Figure 11-(c), the local details are added to the display; 3 As the single-pixel contour edge in Figure 12-(b) will not be clearly displayed in the figure, in order to be able to display the details of the tool contour edge, perform the enlarge display processing; 4. On the basis of ensuring the reasonable layout of the article, the position of the images in the text is fine-tuned, and the image is enlarged as much as possible to be able to display the details clearly.
Point 5: Text should be formatted.
Response 5:
The formatting of the text was checked and revised accordingly based on the reviewers' comments.
Point 6: Is there a possibility to use the proposed research for other topics, classification methods etc.:"Thermographic Fault Diagnosis of Ventilation in BLDC Motors".
Response 6:
Based on the reviewer's suggestion, the research content of the paper was re-summarized, and the title of the paper was revised to reflect the research content of the paper more clearly, and the title was changed to: Machine Vision-based In-situ Detection of Tool Wear in Micro End Milling.
Point 7: Please compare advantages/disadvantages of other approaches etc.
Response 7:
Based on the reviewer's suggestion, an introduction to the indirect method of predicting tool wear status is added to the introduction, firstly explaining its working principle, then discussing the current research status of the method with examples, and finally analyzing the advantages and disadvantages of this method in terms of detection characteristics, implementation cost and difficulty. At the same time, the characteristics and current status of machine vision-based inspection methods are added to further explain the need for the research content carried out in this paper.
Point 8: Please add some sentences about future work.
Response 8:
Based on the reviewer's suggestion, a description of future work was added to the conclusion, including two aspects: 1. improvement of the tool image processing algorithm to improve its accuracy in the detection of wear characteristics; 2. addition of other types of signals in the cutting process in parallel with the acquisition of tool image information, and the artificial intelligence algorithm is used to establish the correspondence between the image and other types of cutting signals, so that the monitoring of the tool wear status is more accurate.
Please also see the attachment.

Round 2
Reviewer 1 Report
Accepted